# Reinforcing intensive motherhood: A study of gender bias in parental responsibilities allocation by large language models

Jiaxing Xiu[1], Yongjie Sun[2,3]*

1 School of Education, Zhengzhou University, Henan, China, 2 Faculty of Arts and Sciences, NYU Shanghai, Shanghai, China, 3 The School of Psychology and Cognitive Science, East China Normal University, Shanghai, China

* ys6261@nyu.edu

## Abstract

This study investigated gender bias in Large Language Models (LLMs) within the context of parenting responsibility attribution, focusing on whether LLMs implicitly reinforce the ideology of "intensive mothering" by assigning caregiving duties predominantly to mothers. Using GPT-4.1 and DeepSeek-V3 as case studies, we used a 3-factor experimental design involving model type, caregiver role (mother, father, or neutral parent), and responsibility framing (prescriptive vs. descriptive). Results revealed an obvious gender bias across both models: mothers were consistently assigned highest caregiving responsibility scores, while fathers received the lowest. Moreover, LLMs produced higher responsibility scores in prescriptive contexts than in descriptive ones, suggesting a tendency to reflect normative social expectations. Mediation analysis showed that gender equality attitudes did not significantly explain these biases, indicating that LLMs' outputs were likely driven by contextual associations in training data rather than consistent ideological positioning. This study extended LLMs bias research into the domain of childrearing, highlighting that even in private contexts, advanced language models tend to reproduce and amplify traditional gender norms. The findings underscored the urgency of incorporating gender sensitivity in LLMs design and training processes. Interventions such as fine-tuning and dataset balancing are essential to prevent these models from reinforcing gendered divisions of labor in parenting.

## Introduction

Debates on parental responsibilities often persisted in gender expectations, ranging from the cultural phenomenon of the "absent father" to the disproportionately high caregiving demands placed on mothers [1]. Mothers continued to shoulder a significantly larger share of parenting duties, including far greater hours of unpaid childcare and household labor compared to fathers [2]. "Intensive Mothering" as an ideology

**Data availability statement:** All research data and materials have been uploaded and are publicly available on the Open Science Framework (OSF) at https://osf.io/nzs49/.

**Funding:** The author(s) received no specific funding for this work.

**Competing interests:** The authors have declared that no competing interests exist.

that is child-centered, expert-guided, emotionally demanding, time and labor intensive, and financially costly, positioning mothers as the primary and self-sacrificing caregivers whose children's needs take precedence over their own [3]. The theory of "Intensive Mothering" believed that mothers bore primary responsibility for child-rearing, emphasizing a child-centered approach in which mothers were expected to dedicate extensive time, energy, and resources to their children, often under expert guidance [3]. This idealized model frames the mother as a wholly devoted, self-sacrificing caregiver, subtly marginalizing the father's role in early parenting [4]. As a result, mothers were subject to intense pressure to meet nearly unattainable parenting standards, while fathers were often exempt from comparable expectations [5].

Building on this, parenting discourses can be understood through the lens of prescriptive advocacy and descriptive advocacy. Prescriptive advocacy refers to normative expectations about what parents *should* do in child-rearing, often reflecting cultural ideals, policy discourses, or expert recommendations that prescribe gendered caregiving roles [6]. Descriptive advocacy, by contrast, reflects how parenting responsibilities are *actually* distributed in daily life, capturing real patterns of involvement rather than idealized norms [7]. This study distinguishes between ideal responsibilities, which refer to socially endorsed parenting standards, and actual responsibilities, which refer to parents' lived caregiving practices. Similarly, the distinction between ideal and actual responsibilities differentiates between socially endorsed standards of parenting (the "ideal") and parents' lived experiences and practices (the "actual"), which may diverge significantly due to structural, cultural, or personal constraints.

Within these social scripts, technological advancement introduced a new dimension for examining how gender roles were either reinforced or challenged. Large Language Models (LLMs), as a groundbreaking technology in the age of artificial intelligence, had developed at an extraordinary pace and are being rapidly integrated into everyday life [8]. LLMs not only revolutionized learning practices but also presented opportunities to empower family education. New-generation first-time parents, particularly during the early parenting stage (1–3 years of the child), lacked prior caregiving experience and had to adapt to sleep and feeding schedules, the physiological demands of breastfeeding, postpartum recovery, and the complex negotiation between professional responsibilities and infant care. These challenges contributed to a common perception of insufficient parenting competence [9].

Recent evidence shows that AI tools are already being used by many parents in concrete ways [10]. Moon et al. [11] conducted focus groups and individual interviews with 28 mothers to explore their perceptions of using Internet resources for parenting. The findings indicated that mothers valued the ability to quickly and anonymously access abundant and diverse information and opinions, while recognizing the importance of relying on reputable sources. They also appreciated the immediacy of support and personalized advice available through social media. Similarly, Quan et al. [12] carried out a quantitative study with 74 U.S. parents of children aged 5–8, revealing that 71.6% of parents had personally interacted with ChatGPT, and 52.7% reported using it for various parenting purposes, including seeking parenting strategies and assisting their children with general and language learning. Chatbot-based

parenting interventions have also been found to be feasible and acceptable in a systematic review [13]. These findings suggest that LLM outputs do not merely exist as hypothetical norms, but are entering into the lived decision-making of many caregivers. Models such as ChatGPT are emerging as significant sources of parenting advice for young parents. Their functionalities–ranging from Q&A-based parenting guidance to the generation of early childhood education content– may be shaping familial role perceptions, including guidance on feeding, educational strategies, and emotional communication [14]. However, the parenting advice offered by LLMs may implicitly carry embedded societal values, including normative assumptions about gender roles. Such biases can reflect and potentially strengthen existing social inequalities [15]. The perceived neutrality of technological outputs may mask the risks of algorithmically amplified gender bias.

Based on this background, the present study investigated whether and how LLMs (represented by ChatGPT and DeepSeek) may reinforce traditional gender roles in parenting by disproportionately assigning primary caregiving responsibilities to mothers. In other words, did these AI models implicitly assume or assert that mothers were the default parent, thereby echoing the logic of intensive mothering while diminishing the father's role? Given the widespread use of chatbots and other LLMs tools in addressing personal inquiries, if the content generated by artificial intelligence perpetuates the notion that child-rearing was primarily "women's work," it may strengthen the very stereotypes that social movements and policy reforms have long sought to dismantle.

While existing studies had identified gender stereotypes in LLMs outputs across domains such as household labor division and career recommendations–for instance, male identifiers were more linked with occupations requiring higher education or physical strength, whereas female identifiers were more frequently associated with roles such as nanny, nurse, or receptionist [16,17]–there remained a gap in research focused specifically on parenting. The central question of this study was whether LLMs implicitly designate mothers as the primary parent, thereby exploring how these technologies may amplify maternal roles while rendering paternal involvement invisible. This inquiry aimed to critically reflect on the role of LLMs in reinforcing existing gender hierarchies. Therefore, the study introduced gender equality attitudes as a mediating variable, using LLMs as a reflective medium through which to magnify prevailing cultural gender scripts. This approach tried to offer a novel perspective on the dynamics of gendered division of childcare in the digital age and, ultimately, to promote more equitable negotiations of parenting responsibilities within families.

## Literature review

### Intensive mothering and the maternal penalty in the context of gender

Theoretical understandings of gender emerged from mid-20th century Western feminist movements [18]. Beauvoir [19], in *The Second Sex*, famously stated that *"one is not born, but rather becomes, a woman."* Gender theory emphasized that gender was not merely a biological distinction but a construct shaped by sociocultural processes [20]. Gendered traits were produced and maintained through social norms, deeply influencing societal structures, power relations, and individual identity formation [21]. Motherhood, in this context, referred to women's social role and identity as mothers within the broader reproductive process [22]. Society often linked motherhood intrinsically with child-rearing–especially care for infants and young children–equating biological functions with socially assigned roles [23]. Hays [3] conceptualized "intensive mothering" as articulating the cultural foundations of these norms., suggesting that a "good mother" had to be wholly devoted, guided by expert advice, and constantly attentive to her child's needs at the expense of personal career and leisure.

Despite growing female participation in the labor market, the responsibilities associated with motherhood–such as marriage, childbirth, and caregiving–continued to exert a negative influence on women's professional advancement [24]. Following childbirth, women invest more time and energy into childcare, which correlates with reduced employment opportunities, higher dismissal rates, and lower wages. Moreover, mothers were perceived as less committed or competent in the workplace due to their caregiving obligations, a phenomenon referred to as the "maternal penalty" [25]. Discrimination against mothers in hiring, compensation, and professional experience had been consistently documented [26,27]. The

persistence of traditional gendered divisions of labor forced women to absorb this penalty, reducing their ability or willingness to invest in their careers. Simultaneously, motherhood tended to reshape women's social interactions into child-centered engagements [28].

The division between maternal and paternal roles was rarely equitable. These distinctions were shaped not only by cultural norms but also by behavioral patterns. Within traditional gender norms around parenting, mothers were viewed as primary caregivers, while fathers were positioned as auxiliary figures [29]. Ideologies of gender portrayed mothers as inherently nurturing and superior caregivers, whereas fathers were stereotypically constructed as financial providers [5]. Regardless of whether the labor was paid or unpaid, mothers consistently dedicated more time to childcare than fathers [24]. Fundamentally, society has been slow to embrace fathers as equal parents and mothers as individuals whose identities extend beyond maternal functions. Current discourses continued to impose normative expectations on women's maternal identities, practices, and experiences, reinforcing constructions of the "good mother." As Heward-Belle [30] argued, women had to be freed from both self-blame and mother-blame. This theoretical trajectory revealed how gender norms reproduced ideological frameworks of intensive mothering, thereby generating structural tensions between maternal identity, professional development, and individual well-being.

## Gender bias in content generated by large language models

Current research had increasingly focused on how Large Language Models (LLMs) manifested bias across different contexts, paying particular attention to the variation in types of biases–such as those related to race, religion, nationality, and gender–depending on linguistic and cultural contexts [31,32]. The origins of bias in LLMs were multifaceted, encompassing the composition of training data, model specifications, algorithmic constraints, product design choices, and policy-level decisions [33]. The underrepresentation of women in AI development and leadership roles might further contribute to the failure of sociotechnical systems to reflect the diverse needs and perspectives of all genders [15]. Ghosh and Caliskan [34] found that when ChatGPT was prompted to replace gender-neutral pronouns with "he" or "she," it tended to reinforce traditional gender stereotypes, such as associating doctors with men and nurses with women, or describing men as going to work while women were cooking or cleaning. A report produced by UNESCO [35] offered a comprehensive analysis of gender stereotyping in LLM-generated content. The findings revealed a consistent pattern in which women were associated with themes such as "home," "family," "children," and "marriage," whereas men were linked with "business," "executives," "income," and "careers".

Beyond these semantic associations, LLMs reproduced and even amplified gender stereotypes present in their training data. Such biases emerged in seemingly neutral outputs [36]. The performative nature of LLMs meant they not only reflected but potentially reinforced exclusionary gender ideologies. Gross [37] noted that ChatGPT exhibited "severe gender bias," reinforcing expectations that women should bear children, nurture and raise them, and make sacrifices for the family, while fathers were rarely portrayed in caregiving or custodial roles. These AI-generated narratives closely mirrored longstanding cultural scripts that framed mothers as primary caregivers and fathers as peripheral supporters, effectively digitalizing the ideology of intensive mothering. Multiple experimental studies supported this claim, demonstrating that LLMs reproduced prevailing societal representations of gender roles–portraying women as nurturers or community members and men as competent or adventurous actors [38]. In doing so, LLMs internalized and regenerated traditional gender roles through their outputs, potentially contributing to the entrenchment of gendered expectations in digital discourse.

## The prescriptive and descriptive dimensions of parental responsibility in LLMs

It has been consistently demonstrated a significant gap between public beliefs about "who should do what" and "who actually does what" in the context of gendered family roles [39]. Previous studies on gender roles revealed a persistent discrepancy between prescriptive expectations of parental responsibility (what ought to be) and descriptive perceptions of its

actual distribution (what is) [40]. The former reflected idealized scripts of how caregiving responsibilities should be divided, whereas the latter emerged from lived, observable practices. This divide between the "prescriptive" and the "descriptive" dimensions of parental role allocation represented a concrete manifestation of the dual nature of social cognition within the family domain. Individuals' judgments about gender-based division of labor were shaped by both normative beliefs (how things should be) and descriptive beliefs (how things are) [41]. As data-driven reflections of societal ideologies [42], LLMs might mirror these tensions in their output logic. Their explicit responses are often aligned with gender-equal values (prescriptive advocacy), while their underlying semantic patterns might subtly perpetuate traditional gendered divisions of labor (descriptive default). For example, when prompted to generate an "ideal model of household task-sharing," LLMs frequently employed progressive discourses of fairness and collaboration [43]. Yet when responding to more context-specific instructions–such as "who typically comforts a baby at night"–they might reinforce the narrative norm that positioned mothers as the primary caregivers [15].

While existing literature had primarily examined these cognitive inconsistencies within human agents, little attention had been given to how such divergences were replicated or encoded within artificial intelligence systems. The present study introduced a framework: prescriptive data captured the LLMs' advocacy stance. The key question was whether the model promoted gender-equal values, especially given descriptive data suggesting that its default logic often treated traditional divisions of labor as natural facts. This study examines how LLMs evaluate maternal and paternal roles in identical caregiving scenarios, in order to reveal whether their gendered assumptions reflect only cultural norms or also reproduce empirical social biases.

### Hypotheses

Building on the above theoretical and empirical background, we designed an experimental framework to examine how Large Language Models (LLMs) allocate caregiving responsibilities across gendered caregiver roles. Specifically, we developed the Infant Care Responsibility Scale, a 69-item measure adapted from four validated caregiver questionnaires aligned with the World Health Organization's five caregiver capability dimensions. Each item was rephrased into a declarative statement representing a specific caregiving behavior (e.g., feeding, sleep regulation, responsive interaction). Both GPT-4.1 and DeepSeek-V3 models were asked to rate, on a 0–100 scale, the extent to which each behavior reflected their perceived caregiving responsibility. Higher scores indicated stronger attribution of caregiving responsibility, meaning that the model considered the caregiver more accountable for performing the described childcare tasks. The resulting mean values constituted the caregiving responsibility scores, which served as the dependent variable. In addition, each model completed the Gender Equality Attitude Scale (GEAS), assessing endorsement of egalitarian versus traditional gender beliefs, which was later included in mediation analyses.

Based on this framework, we formulated the following hypotheses.

H1: LLMs would assign higher caregiving responsibility scores to mothers than to neutral parents, and to neutral parents more than to fathers (mother > neutral > father).

H2: Descriptive judgments (i.e., how responsible the caregiver is) would yield higher scores than normative judgments (i.e., how responsible the caregiver should be).

H3: DeepSeek would exhibit a stronger gender bias than GPT-4o, particularly by assigning significantly higher responsibility scores to mothers than to fathers.

Additionally, we would conduct a mediation analysis of the gender bias on the relationship between caregiver roles and Responsibility scores using PROCESS macro in SPSS.

### Method

#### Pre-registration

This study was pre-registered on the AsPredicted platform (AsPredicted #225975).

## Participants

This study did not involve human participants. Instead, two large language models, GPT-4.1 and DeepSeek-V3, served as the primary subjects of analysis. According to the Artificial Analysis Intelligence Index (AAII) provided by https://artificiala-nalysis.ai/ [44], both models scored 53, indicating a comparable level of artificial intelligence. AAII is a combination metric covering multiple dimensions of intelligence, including MMLU-Pro, GPQA Diamond, Humanity's Last Exam, LiveCode-Bench, SciCode, AIME and MATH-500. Moreover, both models were categorized as non-reasoning models, with GPT-4.1 representing models primarily trained on Western corpora, and DeepSeek-V3 representing those trained on Eastern corpora.

Each model was prompted to respond to all items across each condition, resulting in repeated measures across all factors. Using G*Power, we computed the minimum required sample size for detecting a medium effect ($f = 0.25$) with $\alpha = .05$. The analysis suggested a minimum of 251 total responses. With a 3-factor between-subjects design involving 12 conditions (2 [Model: GPT-4.1 vs. DeepSeek-v3] × 3 [Caregiver role: mother vs, father vs. parent] × 2 [Responsibility construal: Prescriptive vs. Descriptive]), each model produced 25 responses per condition, resulting in 300 responses in total.

We utilized Python scripts to collect model responses via API interfaces. For GPT-4.1, responses were retrieved through the official OpenAI API, specifying "gpt-4.1-2025-04-14" as the model's name. As DeepSeek-V3 is open-sourced and does not support direct API access, we hosted the model using the Novita.ai platform [45], specifying "deepseek/deepseek-V3" in the API parameters. For both models, the temperature was set to 1.0 to balance variability and coherence in the generated responses. In both GPT-4.1 and DeepSeek-V3, the temperature parameter ranges from 0 to 2, where lower values make the output more deterministic and repetitive, while higher values increase randomness and linguistic diversity. A temperature of 1.0 was therefore chosen to maintain a moderate level of variation in the responses while preserving logical consistency and interpretability. All other generation parameters were left at their default values. The system prompt was submitted first, followed by the questionnaire items. All materials are available in Supplementary Material S1.

## Design

A 3-factor between-subjects experimental design was employed, resulting in 12 total conditions. The three independent variables were Model (GPT-4.1 vs. DeepSeek-V3), Caregiver role (mother, father, or gender-unspecified caregiver [parent, control group]), and Responsibility construal (Prescriptive vs. Descriptive).

## Materials

The study utilized two primary instruments: the Infant Care Responsibility Scale and the Gender Equality Attitude Scale(GEAS).

The Infant Care Responsibility Scale was adapted from multiple existing instruments grounded in the five caregiver capability dimensions outlined by the World Health Organization (WHO) in its guideline *Improving Early Childhood Development* [46]. Based on these five domains, we selected four validated scales to measure caregivers' self-assessed caregiving practices (see Table 1). We then adapted the items to reflect responsibility judgments by rephrasing interrogative items into declarative statements. LLMs were asked to rate the degree to which each item represented their caregiving responsibility. The final adapted instrument contained 69 caregiving-related statements, each rated on a 100-point scale, where higher scores indicate stronger perceived responsibility. All adaptations were reviewed and approved by all coauthors and were documented in Supplementary Material S1 File.

The Gender Equality Attitude Scale (GEAS), developed by Chen [51], consisted of 26 items, including 16 reverse-coded items. Responses were recorded using a 7-point Likert scale. Higher scores indicate stronger endorsement of gender-equal beliefs, whereas lower scores reflect greater gender-role bias.

 

**Table 1. Four Scales Corresponding to the Five Caregiver Capability Dimensions.**

| Dimensions | Scale | Reference(s) |
|---|---|---|
| Feeding and Sleep | Caregiver's Nurturing Care Practice Questionnaire (C-NCPQ) | Kong et al. [47] |
| | Infant Sleep Assessment Scales (ISAS) | Feng et al. [48] |
| Responsive Caregiving | Responsive Caregiving Rating Scale (RCRS) | Huang et al. [49] |
| | Caregiver-child Interaction Rating Scale (IRS) | Anme [50] |
| Health and Nutrition | Caregiver's Nurturing Care Practice Questionnaire (C-NCPQ) | Kong et al. [47] |
| | Family Members Participate in Early Education (FMPEE) | Chen and Gong [51] |
| Emergency and Safety Protection | Caregiver's Nurturing Care Practice Questionnaire (C-NCPQ) | Kong et al. [47] |
| | Comprehensive Early Childhood Parenting Questionnaire (CECPAQ) | Verhoeven et al. [52] |
| Early Learning and Family Education | Caregiver's Nurturing Care Practice Questionnaire (C-NCPQ) | Kong et al. [47] |
| | Comprehensive Early Childhood Parenting Questionnaire (CECPAQ) | Verhoeven et al. [52] |
| | Family Members Participate in Early Education (FMPEE) | Chen and Gong [51] |

## Statistical analysis

All statistical analyses were conducted using SPSS version 27.0. We first reported descriptive statistics and correlation coefficients. Subsequently, ANOVAs were performed to examine differences in assigned caregiving responsibilities across models for each variable. Finally, we conducted mediation analyses using the PROCESS macro in SPSS (Model 4; [53]) to test whether gender-role beliefs mediated the relationship between caregiver role and caregiving bias. The analysis employed a bootstrapping strategy with 5,000 resamples to generate bias-corrected 95% confidence intervals for the indirect effects. Indirect effects were considered significant if the confidence intervals did not include zero.

## Results

### Descriptive analyses and correlation

First, we conducted descriptive statistics and correlation analyses. Results showed that the average responsibility score assigned by the LLMs for infant caregiving was 90.55 ($SD = 0.41$). The average score on the Gender Equality Attitude Scale(GEAS) was 6.73 ($SD = 0.14$). Correlation analyses indicated that LLM, caregiver role, and responsibility construal were all significantly associated with caregiving responsibility scores. However, caregiver role and responsibility construal were not significantly correlated with scores on the GEAS. See Table 2 for details.

**Table 2. Descriptive Analyses and Correlation between Variables.**

| | M | SD | 1 | 2 | 3 | 4 | 5 |
|---|---|---|---|---|---|---|---|
| 1 LLM | .50 | .50 | -- | | | | |
| 2 Caregiver Role | 2.00 | .82 | /a | -- | | | |
| 3 Responsibility Construal | .50 | .50 | / a | / a | -- | | |
| 4 Responsibility | 90.55 | 4.12 | .424** | −.125* | .409** | -- | |
| 5 GEAS | 6.73 | 0.14 | −.448** | −.023 | .130* | .070 | -- |

*Note.* a Correlation was not computed because the variables were discrete independent variables. * indicates $p < .05$. ** indicates $p < .01$. *** indicates $p < .001$.

### Hypothesis 1: Differences in caregiver roles

An ANOVA was conducted to test the difference between the caregiver roles assigned to LLMs on the responsibility scores. A significant main effect was found, $F(2, 299) = 19.07$, $p < .001$, $\eta^2 = .114$. Post-hoc analyses were subsequently conducted and adjusted using Bonferroni method. LLMs assigned significantly higher levels of caregiving responsibility under the mother role condition ($M = 92.09$, $SD = 4.44$) compared to the father role condition ($M = 88.72$, $SD = 3.33$), $p < .001$, while the difference between the mother ($M = 92.09$, $SD = 4.44$) and gender-neutral conditions ($M = 90.83$, $SD = 3.83$) was not significant, $p = .07$. In contrast, the responsibility score under the father role condition ($M = 88.72$, $SD = 3.33$) was significantly lower than that under the gender-neutral condition ($M = 90.83$, $SD = 3.83$), $p < .001$. These results suggest the presence of gender bias in LLMs, reflected in the tendency to attribute lower caregiving responsibility to fathers and higher responsibility to mothers. See Fig 1 for details.

### Hypothesis 2: Differences in responsibility construal

Subsequently, we conducted an ANOVA to examine differences in responsibility scores across responsibility construal types. Responsibility construal (Prescriptive vs. Descriptive) and caregiver role were entered as independent variables. Significant main effects were found for both responsibility construal ($F[1, 294] = 68.75$, $p < .001$, $\eta^2 = .19$) and caregiver role ($F[2, 294] = 23.35$, $p < .001$, $\eta^2 = .14$), while the interaction effect was not significant, $p = .63$. Specifically, LLMs assigned higher responsibility scores under the Prescriptive condition ($M = 92.23$, $SD = 3.48$) compared to the Descriptive condition ($M = 88.86$, $SD = 4.04$). See Fig 1 for details.

### Hypothesis 3: Differences in LLMs

To examine differences in responsibility scores across caregiver roles and models, we conducted a two-way ANOVA and tested for an interaction between LLM and caregiver role. The results revealed an insignificant interaction effect, indicating that the two models did not differ significantly in how they assigned responsibility across the three caregiver roles.

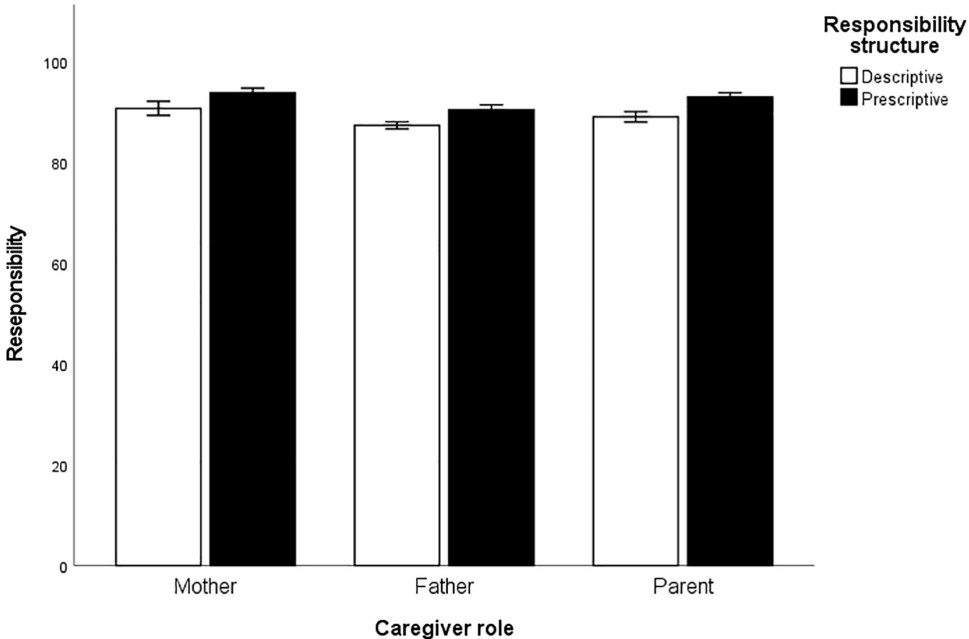

**Fig 1. Responsibility in different Caregiver roles and Responsibility construction.** Note. Error bars represent 95% confidence intervals.

## Additional analyses

A mediation analysis was conducted using Hayes's [53] PROCESS macro (Model 4) to examine whether GEAS mediated the effect of caregiver role (mother vs. father vs. neutral) on responsibility scores. The results indicated that caregiver role significantly predicted responsibility scores (both mother vs. neutral and father vs. neutral comparisons, $ps < .05$). However, gender equality attitudes did not significantly predict responsibility scores ($B = -0.33$, $p = .84$), and the indirect effects of caregiver role on responsibility scores via gender role bias were not significant, as the 95% bootstrap confidence intervals included zero. These findings suggest that gender equality attitude does not mediate the relationship between caregiver role and responsibility attribution. See Fig 2 and Table 3 for details.

## Discussion

### Main Findings

This study focused on the behavior of Large Language Models (LLMs) in contexts involving the distribution of parenting responsibilities and revealed two central patterns. First, the models exhibited gender bias: parenting responsibilities were assigned higher to mothers than to fathers, with the lowest responsibility ratings occurring when gender was unspecified. This pattern could be understood through the lens of gender stereotypes, which encompassed both descriptive beliefs about the behaviors typically exhibited by men and women and prescriptive expectations regarding how each gender ought to behave [7]. Within traditional norms, mothers were expected to serve as primary caregivers, assuming greater responsibility for both household and childcare duties, whereas fathers were expected to prioritize paid work and partici-pate in parenting to a lesser degree [54]. These normative expectations socially legitimized the belief that "mothers should prioritize the family." LLMs, trained on a vast corpora of human language, were likely to internalize these longstanding gendered representations. Cultural depictions of motherhood and fatherhood had been reinforced across books, media, and everyday discourse [55]. Repetition of such gendered associations in language made them appear "true," fostering widespread social acceptance [56]. Consequently, the models did not create gender bias but rather reflected the dominant societal ideologies embedded in their training data. When prompted with parenting-related tasks, the models tended to reproduce an implicit rule that mothers were the primary caregivers. This finding affirmed how social norms were mapped

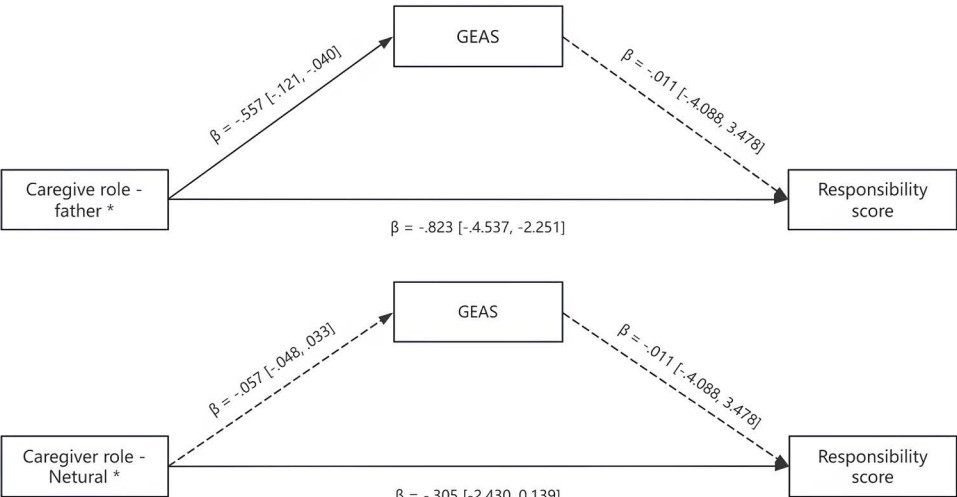

**Fig 2. Mediation Analyses of GEAS between Caregiver Role and Responsibility Score. Note.** * reference level: Caregiver role – Mother; Solid lines indicate significant paths; dashed lines indicate insignificant paths.

**Table 3. Mediation models for caregiver role predicting caregiving responsibility via gender-role beliefs.**

| Path | B | SE | t | p | LLCI | ULCI | β | Significance |
|---|---|---|---|---|---|---|---|---|
| a. Role (Father vs. Mother) → GEAS | −0.08 | 0.02 | −4.05 | <.001 | −0.12 | −0.04 | −.56 | *** |
| b. GEAS → Responsibility | −0.33 | 1.64 | −0.20 | .840 | −3.56 | 2.89 | −.01 | ns |
| c. Total effect (Father vs. Mother → Responsibility) | −3.37 | 0.55 | −6.11 | <.001 | −4.45 | −2.28 | −.82 | *** |
| c'. Direct effect (controlling for GEAS) | −3.39 | 0.57 | −5.98 | <.001 | −4.51 | −2.28 | −.82 | *** |
| ab. Indirect effect via GEAS | 0.03 | 0.15 | — | — | −0.26 | 0.34 | .01 | ns |
| a. Role (Neutral vs. Mother) → GEAS | −0.01 | 0.02 | −0.41 | .680 | −0.05 | 0.03 | −.06 | ns |
| b. GEAS → Responsibility | −0.33 | 1.64 | −0.20 | .840 | −3.56 | 2.89 | −.01 | ns |
| c. Total effect (Neutral vs. Mother → Responsibility) | −1.26 | 0.55 | −2.28 | .023 | −2.34 | −0.17 | −.30 | * |
| c'. Direct effect (controlling for GEAS) | −1.26 | 0.55 | −2.28 | .023 | −2.35 | −0.17 | −.31 | * |
| ab. Indirect effect via GEAS | 0.00 | 0.04 | — | — | −0.07 | 0.11 | .00 | ns |

onto artificial intelligence systems: as Koenig [7] argued, gender stereotypes function as implicit prescriptions for behavior, shaping what was perceived as appropriate conduct for men and women. Based on this, the models' responses were best interpreted not as expressions of independent bias but as computational reflections of prevailing cultural expectations surrounding maternal and paternal roles.

Second, in assessing parenting responsibilities, the models consistently assigned higher scores for prescriptive (ideal) responsibilities than for descriptive (actual) ones. This suggested that when asked to describe ideal conditions, the models attributed greater responsibility to both parents, aligning with societal expectations of moral and normative behavior. In areas like parenting, where morality and social norms were salient, LLMs appeared to favor normative over empirical responses, frequently emphasizing that parents should invest significant time and energy in childcare to fulfill expectations of being "good parents." These mirrored human responses influenced by social desirability: individuals often offered more idealistic evaluations when asked what should be done, while responses about actual behavior tended to be more conservative [7]. In this study, the models might have "assumed" that parents ought to make every effort to care for their children, thereby inflating prescriptive responsibility scores. In contrast, when asked about actual behavior, the models likely relied on their training data–reflecting real-world labor and caregiving divisions, and gave lower descriptive ratings. An additional possibility was that through fine-tuning processes such as Reinforcement Learning from Human Feedback (RLHF), LLMs had learned to provide socially approved responses, especially when addressing normatively charged topics. Therefore, the elevated prescriptive scores might stem from a normative alignment bias. Overall, the observed prescriptive outperformed descriptive pattern illustrated a notable gap between idealized parenting roles and the reality of parental involvement. This discrepancy suggested that LLMs, much like humans, internalized and reproduced the cultural ideal of an "ideal parent" even when such ideals were not consistently reflected in lived experiences.

A comparison between outputs generated by GPT-4.1 and DeepSeek revealed no significant differences in their responses, indicating that both models exhibited similar patterns of gender bias in the context of parenting responsibility allocation. This finding suggested that such biases were not confined to a specific model, but rather represented a shared tendency across mainstream LLMs. This convergence might be attributed, in part, to the fact that leading LLMs were often trained on similarly large-scale corpora that contained analogous cultural information and patterns of bias. Although differences existed in training objectives and architectures across models, they appeared to converge in their internalization of widely distributed gender-role associations. In this sense, gender bias appeared to be a generalizable phenomenon among LLMs, that changing the model does not substantially alter the degree of bias observed. The absence of significant differences between models highlights a critical implication–relying on more advanced or architecturally distinct models

does not automatically mitigate such biases. Instead, there was a clear need for systematic, cross-model debiasing strategies to address these entrenched representational patterns.

Additionally, gender role attitudes did not mediate the observed output bias in either model. This finding indicated that the gender bias present in specific parenting-related contexts was not necessarily linked to the model's broader representations of gender equality. In other words, LLMs may produce stereotypical responses not because of a coherent or consistent "view" on gender roles, but because they responded to context-specific associations derived directly from their training data. For example, a model may appear "progressive" when prompted with general questions about gender equality, but still reinforced traditional norms when asked which parent typically comforted a child. This pattern aligned with research in human cognition, which showed that implicit gender stereotypes such as automatic associations of women with domestic roles, were more predictive of bias outcomes in applied settings (e.g., hiring) than explicit attitudes or self-reports [57]. Correspondingly, the gender bias in LLMs seemed to stem not from a unified "gender role attitude," but from context-triggered implicit learning. In other words, the model retrieved gender-role associations from its training corpus based on the specific cues within a given prompt, and the strength of those associations may vary independently of the model's general positions on gender topics. The non-significant role of gender role attitudes as a mediating variable therefore suggested that model bias could not be fully explained by referencing a single attitudinal dimension. Rather, it called for an exploration into how LLMs internally represented and retrieved knowledge, and how those representations translated into biased outputs under different contextual conditions.

## Implications for AI bias research and gender equality practice

The findings of this study carried important implications for both the research on AI bias and the broader pursuit of gender equality in practice. First, from a research perspective, this study extended the contextual scope of investigations into bias in large language models. Prior researches had primarily focused on biases related to race, occupational stereotypes, or systemic discrimination in public domains. This study discussed the topic toward parenting, a domain rooted in everyday life and deeply saturated with gendered expectations. Our results revealed that even within the seemingly private sphere of family responsibilities, advanced language models reproduced gender stereotypes. This highlighted the need to expand AI bias research beyond public-facing or professional contexts to encompass the nuanced social roles embedded in domestic life, such as the division of caregiving labor.

Second, on a practical level, our findings raised critical concerns about the application of LLMs in domains closely tied to gendered behavior and social expectations. As LLMs are increasingly deployed in educational, advisory, and parenting contexts, responses that implicitly framed mothers as the default caregivers risked reinforcing traditional gender divisions. Because LLMs operated as both producers and disseminators of language, their biased outputs may cloak outdated gender ideologies in a veneer of objectivity and intelligence, making such biases more difficult for users to detect. This reinforcement could lead to the amplification and persistence of gendered norms under the guise of neutrality or expert authority. In the context of parenting, this means that young parents seeking guidance may receive responses that disproportionately assign responsibility to mothers, thereby perpetuating the stereotype of the mother as the primary caregiver. Such outputs had the potential to undermine efforts toward more equitable parenting roles and reinforce existing barriers to paternal involvement. Recognizing and correcting these forms of bias was crucial to preventing the reproduction of gender inequality in the everyday applications of artificial intelligence.

In sum, this study extended research in the literature on LLM bias by directing attention to the domain of family and parenting. It also put forward a concrete challenge for the pursuit of gender equity in technological design: how can we ensure that the development and deployment of LLMs contribute to dismantling, rather than entrenching, traditional gender role divisions? Addressing this question is essential not only for improving AI fairness but also for aligning emerging technologies with the values of a more inclusive and equitable society.

## Limitations and future directions

Despite the meaningful findings of this study, several limitations remained. First, the parenting scenarios and dialogue data used in this research were based on simulated contexts rather than derived from natural interactions. This may limit the ecological validity of the results, as the model's expression of bias in real-world applications could be influenced by the diversity of user inquiries and interaction styles. Moreover, given the increasing number of parents who are already using AI tools to obtain parenting advice, it is important to acknowledge that model-generated content may have begun to shape parental decision-making in real contexts. The current study did not directly examine how such AI-generated advice is applied by users, which limits the generalizability of the findings to authentic parenting practices. Second, while this study compared two models, it did not investigate how training processes and tuning strategies might influence the degree of bias. Factors such as the origin of pretraining corpora, value alignments introduced through reinforcement learning with human feedback, and other fine-tuning strategies may all contribute to shaping model biases, yet were not disaggregated in this study. Finally, there were limitations in how bias was measured. We quantified bias using responsibility score differentials, but it remained uncertain whether this metric sufficiently captures the full extent of gendered tendencies in parenting discourse. Further validation with additional metrics and methods is needed.

To address these limitations, future research may consider incorporating real interaction data by collecting user logs or question sets from actual parenting chatbot usage to examine whether bias expressions in real-world environments are consistent with those found in simulated contexts. Additionally, future studies should consider cross-cultural and multilingual comparisons to evaluate how models trained on different linguistic corpora and serving users from different regions behave in similar parenting scenarios. From an application perspective, future work may explore technical strategies to mitigate gender bias in LLMs, such as introducing debiasing constraints during response generation, post-processing interventions, or fine-tuning models to adopt more neutral representations of parental roles. Experimental interventions–such as rebalancing training data to present fathers and mothers more equally or increasing the presence of positive examples of paternal involvement–could be evaluated to assess their effect on reducing model bias. These efforts would provide empirical support for the development of fairer language models. In the long run, continuous monitoring and correction of bias in more realistic and diverse contexts is essential to ensure that LLMs evolve toward greater fairness and inclusivity, and contribute positively to the advancement of gender equality in society.

## Supporting information

**S1 File. Questionnaire.**
(XLSX)

## Author contributions

**Conceptualization:** Jiaxing Xiu, Yongjie Sun.

**Data curation:** Jiaxing Xiu.

**Formal analysis:** Yongjie Sun.

**Investigation:** Jiaxing Xiu, Yongjie Sun.

**Methodology:** Jiaxing Xiu, Yongjie Sun.

**Project administration:** Jiaxing Xiu.

**Resources:** Jiaxing Xiu.

**Software:** Yongjie Sun.

**Supervision:** Yongjie Sun.

**Validation:** Jiaxing Xiu, Yongjie Sun.

**Visualization:** Yongjie Sun.

**Writing – original draft:** Jiaxing Xiu, Yongjie Sun.

**Writing – review & editing:** Jiaxing Xiu, Yongjie Sun.

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
