## [Decision Letter · Decision Letter 0]

10 Oct 2025

Dear Dr. Sun,

Thank you for submitting your manuscript to PLOS ONE. After careful consideration, we feel that it has merit but does not fully meet PLOS ONE’s publication criteria as it currently stands. Therefore, we invite you to submit a revised version of the manuscript that addresses the points raised during the review process.

We look forward to receiving your revised manuscript.

Kind regards,

Zheng Zhang

Academic Editor

PLOS ONE

Journal Requirements:

4. Please include captions for your Supporting Information files at the end of your manuscript, and update any in-text citations to match accordingly. Please see our Supporting Information guidelines for more information: http://journals.plos.org/plosone/s/supporting-information .

Additional Editor Comments:

Please revise the manuscript carefully in accordance with the reviewers’ comments.

Reviewers' comments:

Reviewer's Responses to Questions

**Comments to the Author**

1. Is the manuscript technically sound, and do the data support the conclusions?

Reviewer #1: Partly

2. Has the statistical analysis been performed appropriately and rigorously?

Reviewer #1: No

3. Have the authors made all data underlying the findings in their manuscript fully available?

Reviewer #1: Yes

4. Is the manuscript presented in an intelligible fashion and written in standard English?

Reviewer #1: No

Reviewer #1: Review of the manuscript “Reinforcing Intensive Motherhood: A Study of Gender Bias in Parental Responsibilities Allocation by Large Language Models”

Context and contribution

Building on the idea that Large Language Models (LLMs) may reproduce social stereotypes, the article investigates whether two frontier systems (GPT‑4.1 and DeepSeek‑V3) implicitly endorse the ideology of intensive mothering -- the culturally dominant expectation that mothers shoulder the bulk of childcare. Using a 3‑factor experiment that crosses (i) model type, (ii) caregiver role (mother, father, neutral parent) and (iii) responsibility framing (prescriptive vs. descriptive), the authors show that both LLMs assign systematically higher caregiving scores to mothers and that prescriptive prompts elicit stronger attributions of responsibility than descriptive prompts. Mediation analysis suggests these effects are not driven by the models’ gender‑equality attitudes but rather by associations embedded in training corpora. By extending bias research into the domestic sphere, the study highlights the risk that advanced language technologies may reinforce traditional gender norms in private life and argues for gender‑sensitive model design and data curation.

1. Major (general) issues

• Missing page numbers: The PDF has no pagination, which makes it difficult for reviewers to reference specific passages. Please insert page numbers before an eventual resubmission.

• Data availability statement: The link provided (https://osf.io/nzs49/?view_only=7746022768c1455bb2a970b14c7d6fcb) leads to an Unauthorized page.

• Citation formatting: The phrase “(Hays, 1996) concept of intensive mothering” needs correction.

2. Specific comments

# Section Comment

2.1 Introduction Several key terms--intensive mothering, prescriptive advocacy, descriptive advocacy, and the distinction between “ideal” vs. “actual” responsibilities--are mentioned but not defined. PLOS ONE has a broad readership; add concise definitions or footnotes.

2.2 Motivation The manuscript implies that parents rely on LLM‑powered tools, but the mechanism connecting model outputs to real childcare decisions remains implicit. Brief statistics on global AI adoption in parenting (e.g., UNESCO 2024) would ground the argument.

2.3 Hypotheses H1 references “caregiving responsibility scores” yet the construction of these scores only reappears deep in the Methods section. Move a short description to the hypothesis paragraph.

2.4 Experimental settings The “temperature” parameter (set to 1.0) is jargon. Add a one‑sentence explanation: e.g., “Temperature controls the randomness of generated text--higher values yield more varied but potentially less coherent responses.”

2.5 PROCESS macro The mediation test is summarized in a single sentence. Readers unfamiliar with Hayes’ PROCESS require more detail: model number, bootstrapping strategy, indirect‑effect confidence intervals, and the operationalization of “gender‑role beliefs.” A short appendix or table would suffice.

2.6 Table/figure clarity Results tables should list exact p‑values or 95 % confidence intervals rather than qualitative descriptors (“obvious gender bias”). Figures illustrating mean scores with error bars would greatly aid interpretation.

3. Minor editing suggestions

• Replace long em‑dashes with the journal’s preferred punctuation or use “--” throughout, for consistency.

• Standardize American vs. British spelling (e.g., “behavior” vs. “behaviour”).

• Check subject‑verb agreement in long sentences (e.g., “LLMs were likely driven” → “LLM outputs were likely driven”).

4. Recommendation

The manuscript addresses a timely and socially significant question and employs a creative experimental design. However, the current draft is conceptually under‑explained and methodologically terse, which obscures its contribution. With clearer definitions, fuller reporting of the mediation procedure, restoration of the data link, and improved formatting, the study has strong potential for publication in PLOS ONE.

I therefore recommend major revisions before the article can be considered for acceptance.

**Do you want your identity to be public for this peer review?** For information about this choice, including consent withdrawal, please see our Privacy Policy

Reviewer #1: No

---

## [Author Response · Author response to Decision Letter 1]

13 Oct 2025

Reviewer #1: Review of the manuscript “Reinforcing Intensive Motherhood: A Study of Gender Bias in Parental Responsibilities Allocation by Large Language Models”

Context and contribution

Building on the idea that Large Language Models (LLMs) may reproduce social stereotypes, the article investigates whether two frontier systems (GPT‑4.1 and DeepSeek‑V3) implicitly endorse the ideology of intensive mothering -- the culturally dominant expectation that mothers shoulder the bulk of childcare. Using a 3‑factor experiment that crosses (i) model type, (ii) caregiver role (mother, father, neutral parent) and (iii) responsibility framing (prescriptive vs. descriptive), the authors show that both LLMs assign systematically higher caregiving scores to mothers and that prescriptive prompts elicit stronger attributions of responsibility than descriptive prompts. Mediation analysis suggests these effects are not driven by the models’ gender‑equality attitudes but rather by associations embedded in training corpora. By extending bias research into the domestic sphere, the study highlights the risk that advanced language technologies may reinforce traditional gender norms in private life and argues for gender‑sensitive model design and data curation.

Thank you very much for your precise summary of our study and for the valuable suggestions you provided below. We have carefully revised our manuscript in response to your comments and have detailed the modifications and their rationale in the following section.

1. Major (general) issues

• Missing page numbers: The PDF has no pagination, which makes it difficult for reviewers to reference specific passages. Please insert page numbers before an eventual resubmission.

Thank you for your suggestion. We have now added precise page and line numbers in the manuscript for your specific reference.

• Data availability statement: The link provided (https://osf.io/nzs49/?view_only=7746022768c1455bb2a970b14c7d6fcb) leads to an Unauthorized page.

Thank you for your suggestion. We also noted that the editor recommended making the data fully accessible during the review stage. Accordingly, we have now made the data publicly available and updated the link to the latest data repository.

All data in this study was available at https://osf.io/nzs49/.

• Citation formatting: The phrase “(Hays, 1996) concept of intensive mothering” needs correction.

Thank you very much for pointing out this error. According to APA 7th edition, in-text citations within a sentence should use the narrative citation format. We have now revised the sentence accordingly.

Hays (1996) conceptualized “intensive mothering” as articulating the cultural foundations of these norms.

2. Specific comments

# Section Comment

2.1 Introduction Several key terms--intensive mothering, prescriptive advocacy, descriptive advocacy, and the distinction between “ideal” vs. “actual” responsibilities--are mentioned but not defined. PLOS ONE has a broad readership; add concise definitions or footnotes.

Thank you very much for your suggestion. In the introduction, we have added definitions of the core concepts of this study, including intensive motherhood, as well as the distinction between the descriptive and the prescriptive dimensions. To avoid excessive overlap with the literature review section, we provided only brief definitions of these key variables in the introduction to enhance the readability of the paper.

“Intensive Mothering” as an ideology that is child-centered, expert-guided, emotionally demanding, time and labor intensive, and financially costly, positioning mothers as the primary and self-sacrificing caregivers whose children’s needs take precedence over their own.

Building on this, parenting discourses can be understood through the lens of prescriptive advocacy and descriptive advocacy. Prescriptive advocacy refers to normative expectations about what parents should do in child-rearing, often reflecting cultural ideals, policy discourses, or expert recommendations that prescribe gendered caregiving roles (Milkie et al., 2002). Descriptive advocacy, by contrast, reflects how parenting responsibilities are actually distributed in daily life, capturing real patterns of involvement rather than idealized norms (Koenig, 2018). This study distinguishes between ideal responsibilities, which refer to socially endorsed parenting standards, and actual responsibilities, which refer to parents’ lived caregiving practices. Similarly, the distinction between ideal and actual responsibilities differentiates between socially endorsed standards of parenting (the “ideal”) and parents’ lived experiences and practices (the “actual”), which may diverge significantly due to structural, cultural, or personal constraints.

2.2 Motivation The manuscript implies that parents rely on LLM‑powered tools, but the mechanism connecting model outputs to real childcare decisions remains implicit. Brief statistics on global AI adoption in parenting (e.g., UNESCO 2024) would ground the argument.

Thank you for your suggestion. We have added a paragraph in the introduction to describe in more detail the current phenomenon of many mothers using AI to obtain parenting advice. Our main argument is that mothers may already be influenced by AI-generated perspectives during the process of seeking advice, even if this influence does not necessarily occur through formal adoption of AI tools. However, few studies have directly examined how mothers apply such AI-generated parenting advice in specific contexts. We acknowledge this limitation and have further elaborated on it in the discussion section.

Recent evidence shows that AI tools are already being used by many parents in concrete ways (Chua et al., 2024). Moon et al. (2019) conducted focus groups and individual interviews with 28 mothers to explore their perceptions of using Internet resources for parenting. The findings indicated that mothers valued the ability to quickly and anonymously access abundant and diverse information and opinions, while recognizing the importance of relying on reputable sources. They also appreciated the immediacy of support and personalized advice available through social media. Similarly, Quan et al. (2024) carried out a quantitative study with 74 U.S. parents of children aged 5 to 8, revealing that 71.6% of parents had personally interacted with ChatGPT, and 52.7% reported using it for various parenting purposes, including seeking parenting strategies and assisting their children with general and language learning. Chatbot-based parenting interventions have also been found to be feasible and acceptable in a systematic review (Klapow et al., 2024). These findings suggest that LLM outputs do not merely exist as hypothetical norms, but are entering into the lived decision-making of many caregivers.

Moreover, given the increasing number of parents who are already using AI tools to obtain parenting advice, it is important to acknowledge that model-generated content may have begun to shape parental decision-making in real contexts. The current study did not directly examine how such AI-generated advice is applied by users, which limits the generalizability of the findings to authentic parenting practices.

2.3 Hypotheses H1 references “caregiving responsibility scores” yet the construction of these scores only reappears deep in the Methods section. Move a short description to the hypothesis paragraph.

Thank you for your suggestion. To enhance the clarity and readability of the manuscript, we have added a description of the overall framework of the current study in the hypotheses section and provided a detailed explanation of the content and meaning of the scale we designed.

Building on the above theoretical and empirical background, we designed an experimental framework to examine how Large Language Models (LLMs) allocate caregiving responsibilities across gendered caregiver roles. Specifically, we developed the Infant Care Responsibility Scale, a 69-item measure adapted from four validated caregiver questionnaires aligned with the World Health Organization’s five caregiver capability dimensions. Each item was rephrased into a declarative statement representing a specific caregiving behavior (e.g., feeding, sleep regulation, responsive interaction). Both GPT-4.1 and DeepSeek-V3 models were asked to rate, on a 0–100 scale, the extent to which each behavior reflected their perceived caregiving responsibility. Higher scores indicated stronger attribution of caregiving responsibility, meaning that the model considered the caregiver more accountable for performing the described childcare tasks. The resulting mean values constituted the caregiving responsibility scores, which served as the dependent variable. In addition, each model completed the Gender Equality Attitude Scale (GEAS), assessing endorsement of egalitarian versus traditional gender beliefs, which was later included in mediation analyses.

2.4 Experimental settings The “temperature” parameter (set to 1.0) is jargon. Add a one‑sentence explanation: e.g., “Temperature controls the randomness of generated text--higher values yield more varied but potentially less coherent responses.”

Thank you for your suggestion. In this section, we have provided a detailed explanation of the meaning and range of the temperature parameter for both models and clarified the rationale for setting the temperature value to 1.

In both GPT-4.1 and DeepSeek-V3, the temperature parameter ranges from 0 to 2, where lower values make the output more deterministic and repetitive, while higher values increase randomness and linguistic diversity. A temperature of 1.0 was therefore chosen to maintain a moderate level of variation in the responses while preserving logical consistency and interpretability.

2.5 PROCESS macro The mediation test is summarized in a single sentence. Readers unfamiliar with Hayes’ PROCESS require more detail: model number, bootstrapping strategy, indirect‑effect confidence intervals, and the operationalization of “gender‑role beliefs.” A short appendix or table would suffice.

Thank you for your suggestion regarding the details of the data analysis. We have now added more technical information about the mediation analysis in the Data Analysis subsection of the Method section. In addition, we have included a supplementary table in the Additional Analyses section of the Results, providing a detailed summary of the mediation analysis results.

2.6 Table/figure clarity Results tables should list exact p‑values or 95 % confidence intervals rather than qualitative descriptors (“obvious gender bias”). Figures illustrating mean scores with error bars would greatly aid interpretation.

Thank you for your suggestion. In the Additional Analyses section, we have added a table presenting detailed results for each path of the mediation analysis, including 95% confidence intervals. We have also revised Figure 1 to include error bars representing the 95% confidence intervals.

3. Minor editing suggestions

• Replace long em‑dashes with the journal’s preferred punctuation or use “--” throughout, for consistency.

Thank you very much for your suggestion. We have replaced all em dashes in the manuscript with en dashes.

• Standardize American vs. British spelling (e.g., “behavior” vs. “behaviour”).

Thank you for your suggestion. We have thoroughly reviewed and revised the manuscript to correct typographical errors and to standardize the language to American English.

• Check subject‑verb agreement in long sentences (e.g., “LLMs were likely driven” → “LLM outputs were likely driven”).

Thank you for your suggestion. We have thoroughly reviewed and revised the manuscript to ensure consistency, including corrections to typographical errors, verb tenses, and voices.

4. Recommendation

The manuscript addresses a timely and socially significant question and employs a creative experimental design. However, the current draft is conceptually under‑explained and methodologically terse, which obscures its contribution. With clearer definitions, fuller reporting of the mediation procedure, restoration of the data link, and improved formatting, the study has strong potential for publication in PLOS ONE.

I therefore recommend major revisions before the article can be considered for acceptance.

We sincerely thank you once again for the opportunity to revise this manuscript. We greatly value your insightful comments and have made multiple revisions throughout the paper in response to your suggestions.

---

## [Editor Report · Decision Letter 1]

16 Oct 2025

Reinforcing Intensive Motherhood: A Study of Gender Bias in Parental Responsibilities Allocation by Large Language Models

PONE-D-25-32416R1

Dear Dr. Yongjie Sun,

We’re pleased to inform you that your manuscript has been judged scientifically suitable for publication and will be formally accepted for publication once it meets all outstanding technical requirements.

Kind regards,

Zheng Zhang

Academic Editor

PLOS ONE

Additional Editor Comments (optional):

The quality of the manuscript has improved after revision; acceptance is recommended.
---

## [Editor Report · Acceptance letter]

PONE-D-25-32416R1

PLOS ONE

Dear Dr. Sun,

I'm pleased to inform you that your manuscript has been deemed suitable for publication in PLOS ONE. Congratulations! Your manuscript is now being handed over to our production team.

Kind regards,

on behalf of

Dr. Zheng Zhang

Academic Editor

PLOS ONE